# Norovirus-mediated translation repression promotes macrophage cell death

Turgut E. Aktepe[1☯¤a], Joshua M. Deerain[1☯¤b], Jennifer L. Hyde[2], Svenja Fritzlar[1], Eleanor M. Mead[1], Julio Carrera Montoya[1], Abderrahman Hachani[1], Jaclyn S. Pearson[3¤c], Peter A. White[4], Jason M. Mackenzie[1] *

1 Department of Microbiology and Immunology, at the Peter Doherty Institute for Infection and Immunity, University of Melbourne, Melbourne, Australia, 2 Department of Microbiology, School of Medicine, University of Washington, Seattle, Washington, United States of America, 3 The Hudson Institute of Medical Research, Centre for Innate Immunity and Infectious Diseases, Melbourne, Australia, 4 School of Biotechnology and Biomolecular Sciences, University of New South Wales, Sydney, Australia

☯ These authors contributed equally to this work.
¤a Current address: Melbourne Veterinary School, Faculty of Veterinary and Agricultural Sciences, The University of Melbourne, Melbourne, Australia
¤b Current address: Australian Centre for Disease Preparedness, East Geelong, Australia
¤c Current address: School of Medicine, University of St Andrews, St Andrews, United Kingdom
* jason.mackenzie@unimelb.edu.au

**Data Availability Statement:** All relevant data within the manuscript can be found here: https://doi.org/10.26188/26332729.

## Abstract

Norovirus infection is characterised by a rapid onset of disease and the development of debilitating symptoms including projectile vomiting and diffuse diarrhoea. Vaccines and antivirals are sorely lacking and developments in these areas are hampered by the lack of an adequate cell culture system to investigate human norovirus replication and pathogenesis. Herein, we describe how the model norovirus, Mouse norovirus (MNV), produces a viral protein, NS3, with the functional capacity to attenuate host protein translation which invokes the activation of cell death via apoptosis. We show that this function of NS3 is conserved between human and mouse viruses and map the protein domain attributable to this function. Our study highlights a critical viral protein that mediates crucial activities during replication, potentially identifying NS3 as a worthy target for antiviral drug development.

## Author summary

Norovirus infection is a global health problem and is it estimated that ~10% of the world population suffer an infection each year. The infection is characterized by projectile vomiting and profuse diarrhoea, yet surprisingly there are no available preventative or treatment options currently available. This is primarily due to the lack of efficient *in vitro* virus cultivation system. Within this report we have used the model norovirus, mouse norovirus or MNV, and investigated the role of the virus encoded protein NS3 in potentiating cell death within infected cells. We show that the initiation of cell death is promoted due to the depletion of cell pro-survival proteins induced by both virus infection and expression of the NS3 protein alone. We have showed that this biochemical characteristic is conserved between mouse and human viruses and mapped the functional domain within the

**Funding:** JMM and PAW were funded by the National Health and Medical Research Council of Australia, grant number 1123135. JMD was supported by a PhD stipend provided by the University of Melbourne and the Miller Foundation. The funders had no role in study design, data collection and analysis, decision to publish, or preparation of the manuscript.

**Competing interests:** The authors have declared that no competing interests exist.

NS3 protein. This research has identified a mechanistic role for the NS3 protein during the norovirus replication cycle.

## Introduction

Noroviruses are capsidated, positive sense single-stranded RNA (+ssRNA) viruses that belong to the *Caliciviridae* family. Although human noroviruses (HuNoV) are highly infectious human pathogens that are the major cause of non-bacterial gastroenteritis cases worldwide, the mechanism of disease is poorly understood [1,2]. Clinically, HuNoV manifestations range from an asymptomatic infection to severe, life-threatening gastroenteritis, and in patients with immune deficiencies can lead to a chronic infection [3]. Globally, 685 million annual infections and over 200,000 annual deaths are linked to HuNoV disease, which leads to an approximate 60 billion USD in losses associated with health care costs and declined productivity [4]. Despite the significant global health burden, effective antiviral treatments and preventative vaccines remain unavailable and are recognized as a public health concern. Progress in developing effective therapies is hampered by the fact that HuNoVs are difficult to cultivate under laboratory conditions, therefore the closely related group V murine norovirus (MNV) acts as a model virus to study HuNoV *in vivo* and *in vitro* [5].

The MNV genome is approximately 7.5 kb, from which the viral proteins are generated via translation of 3 to 4 open reading frames (ORF) and subsequent proteolytic processing by the viral encoded protease (NS6) [6,7]. The genome itself is polyadenylated at the 3' end and the 5' end is covalently linked to the viral protein NS5 (or VPg) which binds select host translation initiation factors to promote efficient replication [8–10]. The remaining non-structural proteins localize to the viral replication complex to aid in viral replication in addition to interacting with host proteins to facilitate replication and regulate cellular homeostasis [11,12]. Of the non-structural proteins, NS3 is of particular importance due to the multifunctional roles it possesses. At the viral level, NS3 localizes to the replication complex and acts as a RNA helicase, chaperone and nucleotide triphosphatase (NTPase) [13,14], whereas at the cellular level, we and others have shown that NS3 associates with the ER, mitochondria, lipid-rich bodies, microtubules and reduces surface expression of MHC-I [15–20]. We, and others, have also demonstrated that MNV infection, via the NS3 protein, arrests host cell cap-dependent translation independent of the integrated stress response and prevents stress granule formation [10,21,22].

Host halting of translation (referred to as host translational shut-off) by viral infection has been well established since it was first discovered during poliovirus infection in the 1960s [23]. Since then, host translational shut-off has been reported with influenza virus (inhibits phosphorylation of eIF2α), alphaviruses such as Chikungunya, Sindbis and Semliki Forest virus (inhibits translation in a PKR-dependent and independent manner) and SARS-CoV-2 (Nsp1 binds to the ribosomal mRNA channel to inhibit translation) [24–28]. Host translational shut-off is a major host defence mechanism against invading viruses. Since viruses lack their own translational machinery, they are completely dependent on the host to translate their genome. Therefore, viruses must orchestrate translational shut-off in a timely manner for favourable access to host-cell machinery and to promote viral replication. During infection, MNV induces translational shut-off 6 hours post infection. This provides sufficient time for MNV to construct its replication complex and establish exponential replication [21], however the exact mechanism of how MNV halts host translation is poorly understood. It has been shown that the viral protease (NS6) may cleave the translation factor PABP to promote a translation bias

in infected cells [9] and infection stimulates the MAPK pathway to activate eIF4E phosphory-lation [29]. We and others have shown that during infection translation repression is uncou-pled from PKR activation [21,22] and thus occurs via an unknown mechanism.

Apoptosis is a key form of programmed cell death which has been extensively studied in the context of viral infections. In general, our cells produce pro-survival proteins particularly from the B-cell lymphoma 2 (BCL-2) family of proteins, including Myeloid cell leukemia 1 (MCL-1), BCL-XL and BCL2, that prevent apoptosis and maintain cellular homeostasis. The induc-tion of cell death itself involves a multitude of signalling and activation pathways with one of the final stages involving the cleavage of the poly ADP-ribose polymerase (PARP) protein resulting in significant DNA damage. While apoptosis is part of the innate immune response and important for control of some pathogens, in others, it can perform a pro-viral function. We and others have described intrinsic apoptosis during norovirus infection and shown that caspase 3-mediated apoptosis is required for efficient viral replication and the proteolytic pro-cessing of the NS1/2 proteins [30–34]. The mechanisms driving apoptosis in MNV-infected cells is not clearly understood. Published reports have shown MNV downregulates the inhibi-tor of apoptosis (IAP) protein survivin and proposed cathepsin B as a non-canonical mecha-nism for induction of apoptosis [30,32]. Additionally, expression of the full ORF1 polyprotein was found to induce apoptosis but attempts to identify a specific viral protein were unsuccess-ful [33]. Recently, it was shown that the N-terminus of MNV NS3 protein may act as a MLKL-like protein to induce pores in the mitochondrial membrane and promote cell death [35].

Based on the importance of host translational shut-off by MNV infection, we were inter-ested in identifying the viral mechanism involved in regulating this process and given the established link between translational repression and apoptosis in other pathogens, we sought to establish if MCL-1 loss was involved in driving apoptosis observed during MNV infection. In this study, we demonstrate that the MNV and HuNoV non-structural protein NS3 alone are responsible for host translational shut-off, independent of other viral factors and are suffi-cient to induce apoptotic cell death through a previously unrecognised mechanism. By adopt-ing the Alanine scanning method, we reveal the key nucleotides within NS3 that are responsible for this process.

## Materials and methods

### Cells and virus infection

Immortalised bone marrow-derived macrophages (iBMDM), Hela and HEK 293T cells were maintained in Dulbecco's Modified Eagle's Medium (DMEM) (Gibco) supplemented with 10% foetal calf serum (FCS) (Gibco) and 1% GlutaMAX (200mM) (Gibco). All cell lines were cultivated at 37˚C in a 5% $CO_2$ incubator. For infection iBMDM cells cultured to 80% con-fluency in a 24 well plate were infected with MNV (strain CW1) at MOI 5 [36]. At 3-hour intervals between 9 and 24 hours post infection, cell supernatant was collected clarified by low-speed centrifugation and stored at -80oC.

### Plasmid preparation

Plasmids encoding the 6xHis-tagged MNV non-structural proteins (NS1-2, NS3, NS4, NS5, NS6, NS7) on a pcDNA3.1 backbone have been generated and published previously [37]. NS3 truncation mutants were constructed by amplifying each fragment with a 5' *XhoI* site and a 3' *BamHI* site incorporated into the primer pairs (File Table 1 at https://doi.org/10.26188/26332729). Each fragment was amplified by PCR using the Q5 High-Fidelity DNA Polymerase (NEB, Cat #: M4091L) following the manufacturers protocols. Each PCR product was digested with *XhoI* and *BamHI* and ligated with the T4 DNA ligase (Promega, Cat #: M1794) into a

*XhoI* and *BamHI* pre-digested pcDNA3.1-mCherry-HIS vector. Full length NS3 triple Alanine mutants (if an alanine was present, this was mutated to a Glycine) were constructed using site-directed mutagenesis in the pcDNA3.1-NS3-mCherry-HIS plasmid. PCR amplification of pcDNA3.1-NS3-mCherry-HIS plasmid was done by using PfuUltra HotStart DNA Polymerase (Agilent, Cat #: 600390) and forward and reverse primers (File Table 2 at https://doi.org/10.26188/26332729) containing site-specific mutations following the manufacturers protocol.

## Chemicals and antibodies

The following antibodies have been used: Guinea Pig anti-MNV NS3 was kindly provided by Kim Green and used for the immune-electron microscopy; Mouse anti-BCL-XL (CST Cat #: 2764); Mouse anti-BCL-2 (CST Cat #: 2875); Rabbit anti-MCL-1 (CST Cat #: 5453); Rabbit anti-PARP (FL and Cleaved) (CST Cat #: 9542L); Rabbit anti-Cleaved Caspase-3 (Asp175) (5A1E) (CST Cat #: 9664S); Rabbit Anti-Actin Affinity Isolated (Sigma, Cat #: A2066-0.2ml); Mouse Anti-Puromycin [3RH11] (Kerafast, Cat #: EQ0001); Rabbit Anti-mCherry (Abcam, Cat #: ab183628); Rabbit Anti-Calnexin (Abcam Cat #: ab22595); Rabbit Anti-6X His tag antibody—ChIP Grade (Abcam, Cat #: ab9108).

The following chemicals were used: Pan Caspase OPH Inhibitor Q-VD, Non-omethylated (QVD) (R&D systems Cat #: RDSOPH00101M) diluted to a final concentration of 10μM in DMSO; MG-132 Ready Made Solution (Sigma-Aldrich, Cat # M7449) diluted to a final concentration of 0.5μM in DMSO; Dimethyl sulfoxide (DMSO) (Sigma-Aldrich, Cat # D8418) was used a vehicle control; Puromycin HCl (Sigma-Aldrich, Cat # P8833) was added to cells at a concentration of 10 μg/ml at indicated times prior to cell lysate collection.

## Lipofectamine 3000 transfections

Seeded cells were incubated until 80% confluence. 1 μg of DNA, 2 μl of P3000 reagent in 23 μl of Opti-MEM (Gibco); and 1 μl Lipofectamine 3000 (Life Technologies) in 24 μl Opti-MEM were incubated at RT for 2 mins. DNA and Lipofectamine mixtures were combined and incubated for a further 10 mins. Meanwhile, cells were washed with fresh cell culture media. 400 μl of cell culture media, containing chemicals where described, was added to each well. On top, the DNA:Lipofectamine mixture is added drop wise. Cells were incubated at 37°C until required. Method is for a 24-well plate. Protocol is scaled according to used plate.

## MNV plaque assays

For plaque assays, 1:10 serial dilutions of cell supernatants were prepared in DMEM and 6 dilutions ($10^{-2}$–$10^{-7}$) were used as inoculum in duplicate. RAW264.7 cells, seeded for 70% confluency on the previous day, were infected in duplicate with the diluted supernatants for 1 hour. Infected RAW264.7 cells were cultured for 48hours with overlay media (70% DMEM, 2.5% [vol/vol] FCS, 13.3 mM NaHCO3, 22.4 mM HEPES, 200 mM GlutaMAX, and 0.35% [wt/vol] low-melting-point agarose) and plaques visualised by fixing with 10% formalin for 1 hour and staining with toluidine blue.

## LDH release cell viability assay

Cell viability of MNV infected iBMDM cells was assessed using CytoTox 96 Non-Radioactive Cytotoxicity assay Promega according to manufacturer's instructions. Briefly, 50uL of cell supernatant was collected from MNV-infected cells every 3 hours between 9 and 24 hours post infection. Cell supernatant was incubated with 50uL CytoTox 96 reagent in a flat-bottom 96-well plate for 20 minutes before reaction stopped and absorbance read at 490nm.

## Western blots

Lysates from MNV-infected or NS3-transfected cells were harvested on ice for 30mins in KALB lysis buffer [150 mM NaCl, 50 mM Tris-HCl, pH 7.5, 1% (v/v) Triton X-100, 1 mM EDTA] supplemented with 1% Protease Inhibitor cocktail III. Samples were centrifuged to isolate the soluble fraction, which was diluted in laemmli sample buffer, heated to 90°C for 10 minutes and equal volumes loaded into a 4–12% polyacrylamide gels. Proteins were separated by SDS-PAGE, transferred to PVDF a membrane, and blocked with 5% skim milk powder in TBS-T (TBS plus Tween). Primary antibodies were prepared in 5% BSA/TBS-T and incubated with membrane overnight at 4°C. The following day, primary antibodies were removed, membrane washed three times with TBS-T and secondary antibodies added for 2 hours at room temperature. Visualisation was performed using Amersham ECL Western Blotting Detection Reagent or Western Lightning Ultra (Perkin-Elmer) on the GE Healthcare Life Sciences AI600 Imager.

## Immunofluorescence

Cells were rinsed twice with Phosphate buffered saline (PBS) and fixed 4% v/v paraformaldehyde (PFA)/PBS for 15 min at RT. Fixative was removed and cells were permeabilised with 0.1% v/v Triton X-100 for 10 min at RT. Cells were rinsed twice with PBS and quenched with 0.2 M glycine for 10 mins at RT. Cells were then rinsed with PBS and coverslips were incubated in primary antibodies diluted in 25 μl of 1% bovine serum albumin (BSA)/PBS for 1 hr at RT. Following incubation with primary antibodies, cells were washed thrice with 0.1% BSA/PBS. Coverslips were incubated in secondary antibodies diluted in 25 μl of 1% BSA/PBS for 45 min at RT. Cells were washed twice with PBS and incubated for 5 mins with 4,6-diamidino-2-phenylindole (DAPI) (0.33 μg/ml) in PBS. Coverslips were rinsed twice with PBS and MilliQ water and mounted on cover-slides with ProLong Diamond (Life Technologies). Cells were analysed using the Zeiss LSM710 confocal microscope.

## Brightfield microscopy

iBMDM cells cultured in a 24-well plate and infected with MNV or left uninfected according to procedure described above. At 24 hours post infection the integrity of the cell monolayer was observed under 5X magnification with DMI4000B Automated Inverted Microscope (Leica Microsystems).

## Electron microscopy

Methods for cryofixation, preparation of cryosections and immunolabelling with anti-NS3 antibodies have been described previously [17]. The sections were then viewed on a JOEL 1010 transmission electron microscope and images were captured on a MegaView III side-mounted CCD camera (Soft Imaging Systems, USA) and processed for publication in Adobe Photoshop.

## Flow cytometry

For fixation, cells were centrifuged at 400xg for 3 mins, washed twice with PBS and resuspended in a BD Cytofix/Perm buffer (Cat #: 51-2090KZ) for 15 mins at RT. Cells were centrifuged and the fixation buffer was aspirated. Cells were washed twice in BD Perm/Wash (Cat #: 51-2091KZ) and resuspended in BD Perm/Wash containing antibodies. Samples were incubated at 4°C in the dark for 30 mins followed by 3 washes with BD Perm/Wash. If unconjugated antibodies were used, the previous steps were repeated with the secondary antibodies

containing fluorophores. Cells were resuspended in PBS and kept in the dark at 4˚C until flow cytometry analysis. Flow cytometry data were collected with a BD LSR Fortessa analyzer using BD FACS Diva software (BD Biosciences). Data were analyzed using FlowJo analysis software.

## Results

### iBMDM cells undergo a rapid loss of cell viability over the course of MNV infection that correlates with infectious virus release

In a recent study, we showed that intrinsic apoptosis was induced in response to MNV infection and was a requirement for efficient viral replication [31]. As a continuation of this study, we further aimed to investigate the mechanisms driving programmed cell death during infection. In support of our previous findings, we initially looked to assess the kinetics of cell death and MNV production over the course of a 24-hour infection. To determine the production of infectious virus, we infected iBMDM cells with MNV at an MOI of 5 and collected supernatants containing virus for enumeration by plaque assay at 9, 12, 15, 18, 21 and 24 hours post infection. The same supernatant was also used to measure cell viability using the CytoTox 96 Non-Radioactive Cytotoxicity assay kit. We observed that cell viability rapidly decreased from ~9 hours post-infection and that almost complete loss of cell viability was observed at ~18 h.p.i (Fig 1A, red line). This virus-induced cytopathic effect could be also easily visualised by light microscopy at 24 h.p.i with the appearance of cell blebbing representing apoptotic cell death (Fig 1B). Interestingly, the reduction in cell viability was found to be closely associated with the production and accumulation of infectious virus in the cell supernatant. Between 12 and 18 hours post infection we observed an exponential increase in virus produced and a greater than 50% decrease in cell viability (Fig 1A, blue line).

From these observations we suggest that there is a link between cell viability and infectious virus release. However, we note that even when cell viability is still reasonably intact (*i.e.* at 9 hours post infection) there is a significant recovery of infectious virus in the supernatant. Thus, it would appear that there are multiple virus egress and release pathways that include cell lysis and release in extracellular vesicles as proposed by others [38,39].

### MNV infection leads to cell death and the loss of pro-survival BCL-2 family proteins coinciding with induction of apoptosis

As we have shown that MNV infection leads to rapid loss of cell viability and demonstrated that apoptosis is the primary mechanism of MNV-induced programmed cell death, we aimed to investigate the involvement and/or contribution of key host proteins involved in the apoptotic pathway. Cell lysates were collected from infected and uninfected cells at 0, 9, 12, 15, 18, 21 and 24 hours post infection, and the abundance of known protein markers of apoptosis (namely capsase-3 and PARP) was assessed by western blotting. Coinciding with the induction of cell death (Fig 1A and 1B), we observed a clear indication of cleavage products of caspase 3 and PARP from 15 hours post infection onwards (Fig 1C). The presence of these cleavage products supports previous reports and suggest apoptosis as the mechanism of MNV-induced programmed cell death [6,30,33].

Whilst investigating the mechanism of cell death during MNV infection, we also immuno-blotted and quantified the relative levels of the BCL-2 family proteins: MCL-1, BCL-2 and BCL-XL. These proteins play an essential role in regulating cell death by inhibiting the pro-apoptotic proteins BAX and BAK, thereby preventing the induction of apoptosis. In our analyses we observed a significant decrease in each of these proteins over the course of a 24-hour infection, proceeding from 15 hours onwards (Fig 1C and 1D). Of

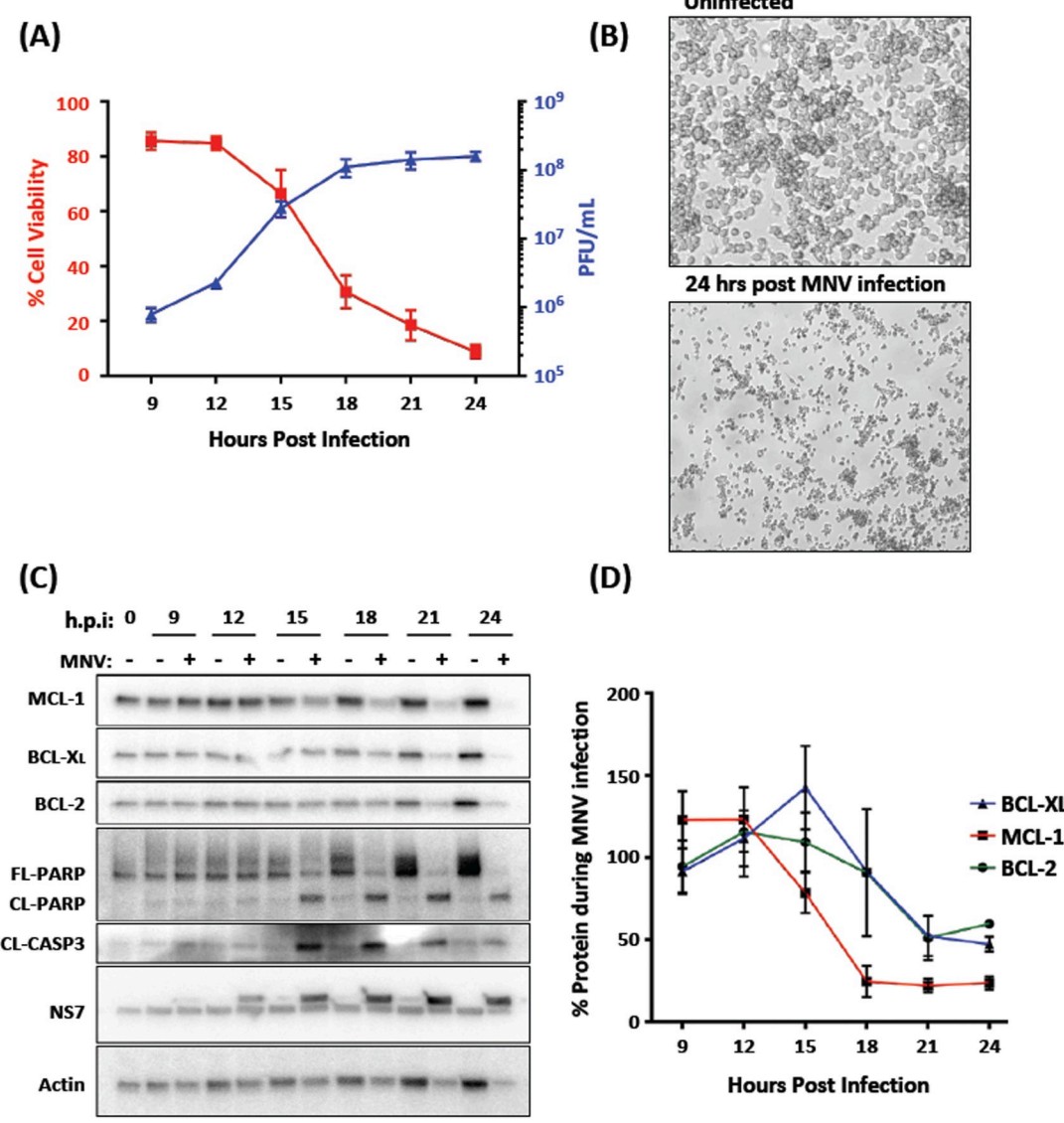

**Fig 1. MNV infection leads to apoptosis and loss of cell viability in iBMDMs.** (A) iBMDMs were infected with MNV (MOI 5) and supernatant was collected at 3-hour intervals between 9- and 24- hours post infection. Infectious virus was quantified for each time point by plaque assay and plotted on the right y-axis (Blue line; n = 2; SEM). LDH release assay to calculate % cell viability relative to total cell lysis control was performed on each time point and plotted on the left y-axis (Red line; n = 2; SEM). (B) iBMDMs infected with MNV (MOI 5) or left uninfected were observed under 5x magnification at 24 hours post infection. (C) Lysates from iBMDMs infected with MNV (MOI 5) were collected at the indicated times and immuno-blotted for markers of apoptosis, cleaved (CL)-caspase- 3 and poly(ADP-ribose) polymerase (Full-length [FL] and cleaved [CL]-PARP, along with pro-survival BCL-2 family proteins, MCL-1, BCL-XL, BCL-2. (D) The relative levels of pro-survival proteins compared with 0 h. p.i were estimated from immuno-blots using image-based quantification (n = 2; SEM).

specific note was the rapid loss of MCL-1, coinciding with activation of apoptosis at 15 hours post infection (Fig 1C and 1D). MCL-1 is a key controller of intrinsic apoptosis and reported to be rapidly turned-over through proteasomal degradation. Our findings are in support of other findings which have shown loss of MCL-1 as a driver of intrinsic apoptosis during infection [40–44], but are the first observations that MNV infection results in a dramatic reduction of this protein.

## The MNV NS3 protein induces apoptosis and MCL-1-loss through host-cell translation repression

Our previous work has shown a role for the MNV NS3 protein in modulating innate immune signalling through repression of host-cell translation [21]. Given that MCL-1 regulation of apoptosis requires frequent turn-over of the protein, we hypothesised that the NS3 protein may also be responsible for the loss during infection. To assess this, the MNV nonstructural (ORF1) proteins were individually expressed in 293T cells either treated with puromycin or left untreated and lysates collected to assess active translation and apoptotic markers. Immuno-blotting for puromycin incorporated into translating proteins is used to measure active protein synthesis in cells [21,45]. Over-expression of NS3 was shown to completely halt host cell translation (Fig 2A) and in support of previously published results [21]. Over-expression of NS3 was also sufficient to substantially decrease the levels of MCL-1 compared to mock transfected cells and cells transfected with the other MNV nonstructural proteins (Fig 2B). Furthermore, we observed that a significant proportion of PARP was cleaved in NS3 transfected cells indicating apoptosis activation. To a lesser degree, PARP cleavage was also observed in NS1-2 transfected cells, however this was not associated with a reduction of MCL-1 nor translational repression suggesting NS1-2 may induce apoptosis via a different mechanism (Fig 2B). Intriguingly, we did observe that co-expression of NS1-2 with NS3 did exacerbate lytic cell death but the mechanism of this is unclear and was not part of this investigation (File Fig 3 at https://doi.org/10.26188/26332729). The repression of translation (Fig 2A), loss of MCL-1 and cleavage of PARP (Fig 2B) in only the NS3 transfected samples leads us to propose that NS3 is responsible for inducing apoptosis in response to infection.

To further validate these findings and determine whether translational repression or apoptosis is the initial step induced by NS3, we transfected cells with NS3, and treated these cells with or without the pan-caspase inhibitor QVD or the proteosome inhibitor MG132 and performed the Puromycin incorporation assay. Significantly, we observed QVD was sufficient to prevent the NS3-induced apoptosis in NS3 transfected cells but did not restore host-cell translational shut off (Fig 2C). This result indicated that our observed translation repression was not dependent on apoptosis activation. Similarly, QVD did not protect against MCL-1 loss in NS3-transfected cells suggesting that MCL-1 reduction is not dependent on apoptosis (Fig 2D, ii). However, strikingly the MCL-1 levels were rescued when NS3-transfected cells were treated with MG-132 (Fig 2D, iii). Proteosome inhibition prevents the rapid turn-over of MCL-1 and more significantly prevented the induction of apoptosis in NS3-transfected cells, observed by the prevention of PARP cleavage.

Taken together, these results indicated that the MNV NS3 protein attenuates host protein synthesis that results in a depletion of the essential pro-survival protein MCL-1, which in turn triggers apoptosis induction.

## The MNV NS3 protein is confined to the viral replication complex during infection of murine macrophages

Recently, it has been reported that the MNV NS3 protein contains an N-terminal MLKL-like domain that inserts into the mitochondrial membrane to induce pores and ultimately cell death [35]. Previously, we had investigated the intracellular localisation of the MNV NS3 and shown that it was resident within the viral replication complex (RC) with viral dsRNA [11,17]. Thus, we investigated the distribution of NS3 again but focussed more on any potential mitochondrial localisation over the course of infection using immunogold labelling (Fig 3). As discovered previously, we could clearly observe significant labelling with anti-NS3 antibodies in

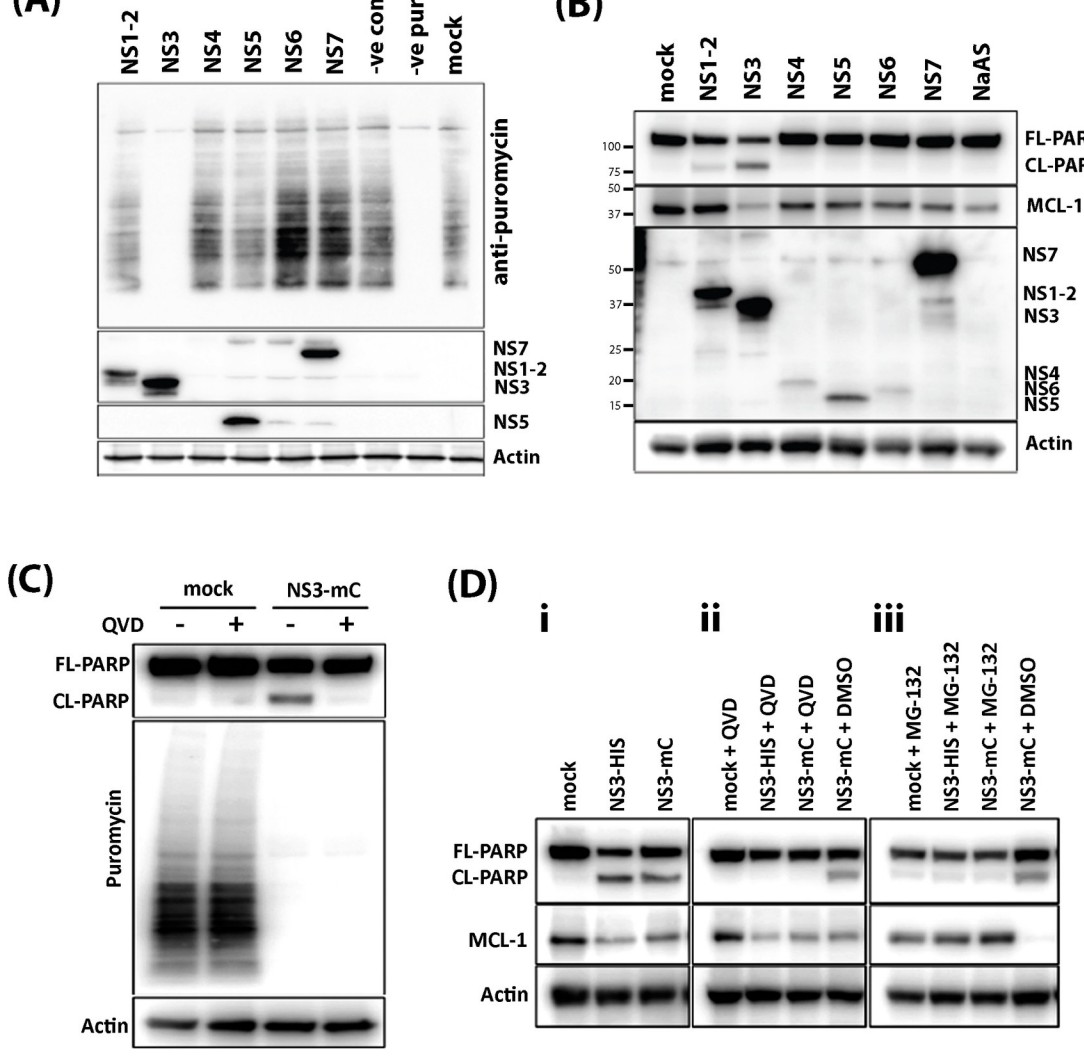

**Fig 2. MNV NS3 protein induces translational shut-off and apoptosis.** (A) A puromycin incorporation assay was assessed on lysates harvested from 293T cells transfected with expression plasmids encoding MNV nonstructural proteins fused with His tag or left untransfected. MNV nonstructural proteins were observed by staining with Anti-6X His tag antibody. The NS4 and NS6 proteins could not be observed by western blot on this occasion but can be observed in Panel (B). NaAs is sodium azide and was used to invoke a stress response in cells. Host cell translation was measured by treating with puromycin for 30 mins and measuring incorporation with anti-puromycin antibody. Mock represents cells not transfected but pulsed with puromycin, -puro represents mock-transfected cells without addition of puromycin and -ve control are cells transfected with the control plasmid and pulsed with puromycin. (B) Immuno-blot of lysates harvested from 293T cells transfected with expression plasmids encoding MNV nonstructural proteins fused with His tag. The membrane was probed with anti-6X His tag, PARP, MCL-1 and actin antibodies. Proteins were visualised with species-specific IgG conjugated to HRP and ECL. (C) Immuno-blot of lysates collected from cells transfected with expression plasmid encoding NS3 fused with mCherry (NS3-mC), treated with caspase inhibitor QVD throughout transfection or left untreated and pulsed with puromycin as above. (D) Immuno-blot of lysates from 293T cells transfected with expression plasmid encoding NS3 fused with His tag (NS3-His), mCherry (NS3-mC) or left untransfected. Cell treated were either untreated (i), treated with caspase inhibitor QVD (ii), or treated with proteosome inhibitor MG-132 (iii) throughout transfection. The membrane was incubated with antibodies specific for PARP, MCL-1 and actin.

the virus RC at all stages of infection. NS3 was observed to specifically localise to the membrane of the RC at all time points investigated (Fig 3A–3C). However, we did not observe any significant labelling of the internal membrane of the mitochondria nor the outer membrane. In addition, the morphology of the mitochondria appeared visibly normal.

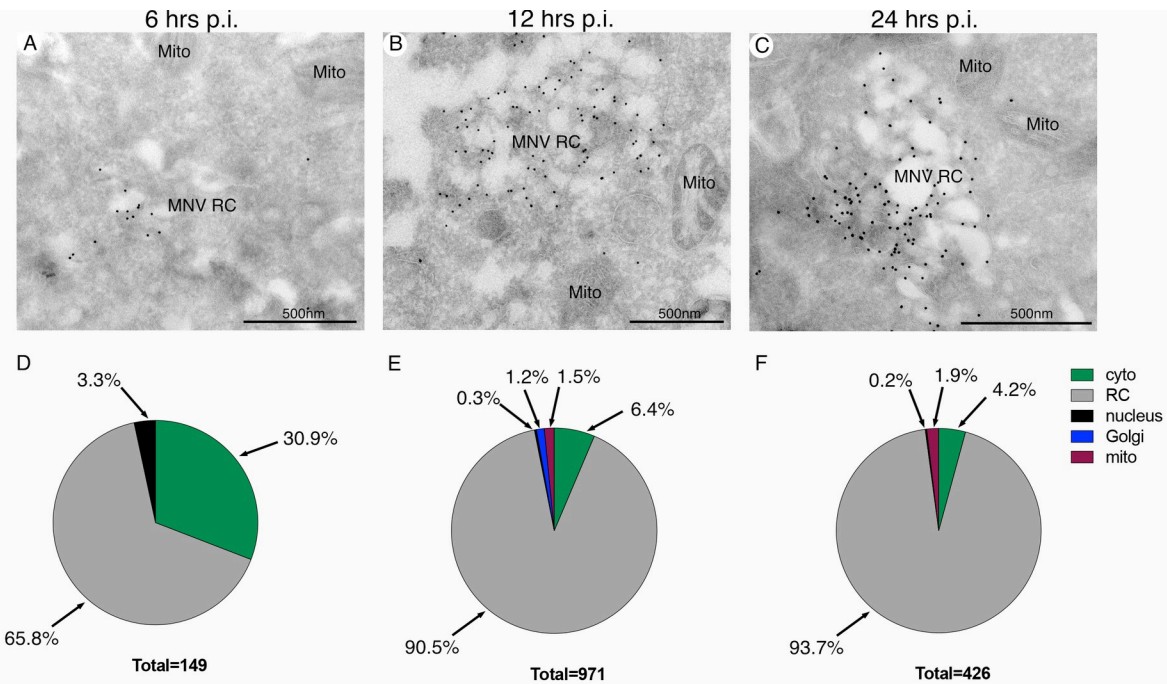

**Fig 3. MNV NS3 localises predominantly to the viral replication complex during infection.** RAW264.7 cells were infected with MNV (MOI 5) and at 6 (A), 12 (B) or 24 (C) hrs p.i. the cells were collected and processed for cryosectioning and immunogold labelling with anti-NS3 antibodies and 10nm Protein-A gold. (D-F) Quantitation of the anti-NS3 labelling on defined cellular organelles at each time point as specified. Cytoplasmic localisation was defined as any gold particle observed not associated with a membrane or organelle. The MNV replication complex (RC) and Mitochondria (Mito) are highlighted. Magnification bars in all panels are 500nm.

Thus, we would conclude that the vast majority of the intracellular NS3 produced during infection is localised and contained within the viral RC. Arguably little to no NS3 could be observed associated with the mitochondria at the ultrastructural level.

## Localization and intracellular distribution of mouse norovirus NS3 protein is dependent on the expression of different domains of the protein

To determine the region responsible for the NS3-imposed translational shut off, the 364 amino acid NS3 protein was segmented into 8 different regions based on the three known domains; N-term (amino acids 1 to 105), core (amino acids 130 to 285) and C-term (amino acids 290 to 364). We thus generated truncated mutants of the NS3 protein by recombinantly expressed the amino acid regions of NS3 1 to 34, 1 to 67, 1 to 100, 1 to 134, 1 to 182, 135 to 300, 183 to 364 and 301 to 364. To aid in the visualisation and identification of the expressed protein each mutant was fused to the fluorescent protein mCherry (mC) at the C-terminus (Fig 4A).

We initially determined the production and localisation of the expressed mutants by IFA (Fig 4B). We observed that the full length NS3-mC localised diffusely throughout the cytoplasm but also to distinct cytoplasmic foci we had observed previously [17]. These foci were also observed with the NS3-mC(1–182), (135–300) and to some extent the (1–134) constructs. Noting that like we have observed during expression of the full-length protein [15–17], these distinct foci are not observed in all cells but are combined with the more ER-like cytoplasmic staining. Notably removal of the C-terminus resulted in a more diffuse, ER-like distribution as observed with NS3-mC(1–67), (1–100), and (1–134). Conversely, removal of the N-terminus resulted in a more non-specific diffuse localisation pattern, as observed with NS3-mC(135–

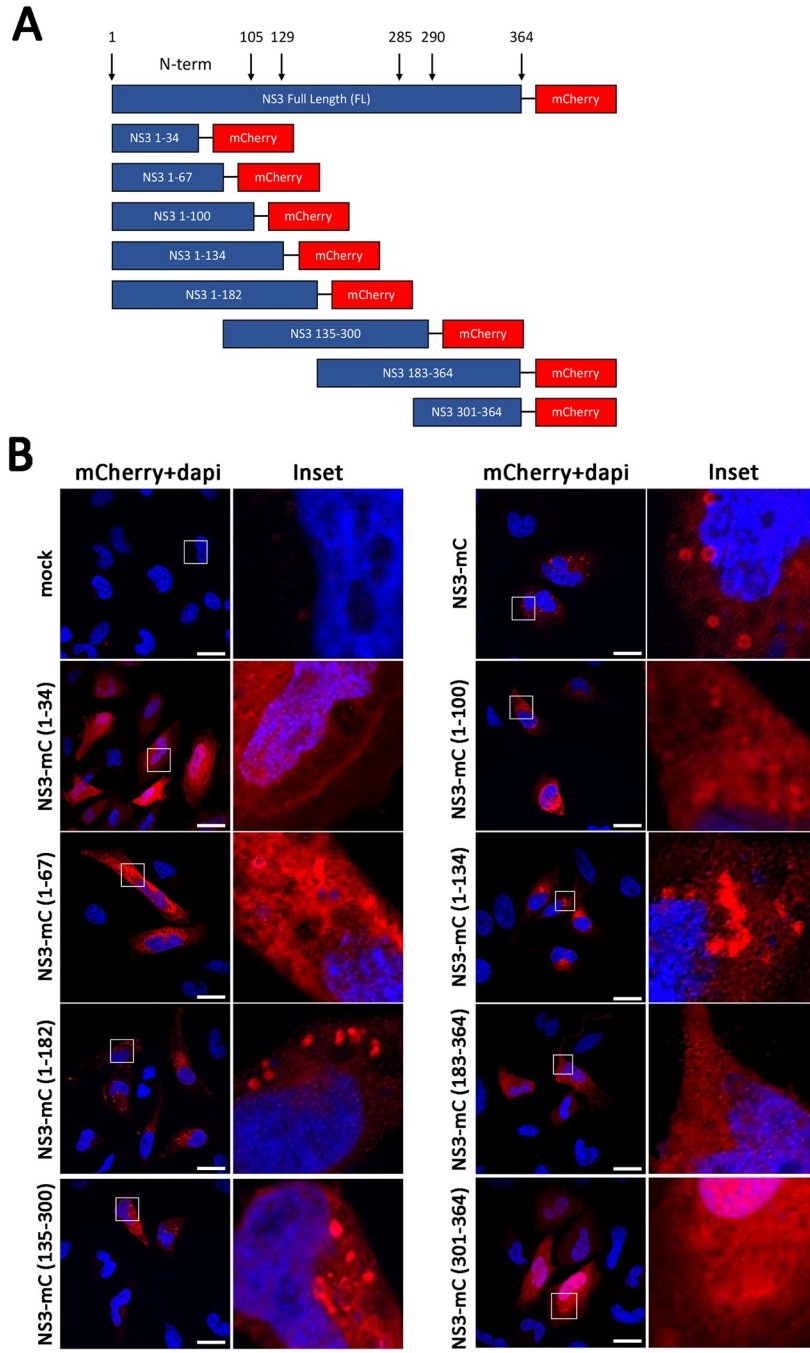

**Fig 4. Cellular localisation of MNV truncation mutants.** (A) Schematic of NS3 truncation mutants generated in pcDNA3.1 expression constructs with key domains illustrated. Arrows indicate the amino acid positions of the three NS3 domains. (B) Immunofluorescent staining of HeLa cells transfected with mCherry-tagged NS3 expression constructs (white) and fixed and permeabilised before counterstaining with DAPI (blue). Images were captured using a Zeiss LSM 710 confocal microscope and analysed with ZEN software.

300) and (301–364). Intriguingly, the minimal N-terminus construct, NS3-mC(1–34), displayed a very tight plasma membrane banding-like pattern but still appeared to be membrane associated. We additionally, assessed any localisation of the NS3 truncation mutants with mitochondria by co-staining cells with MitoTracker Deep Red. Overall, minimal co-

localisation was observed and if present, co-localisation co-efficient was significantly different when compared to the known mitochondrial protein, MAVS (File Fig 2 at https://doi.org/10.26188/26332729).

Overall, these observations indicate that the individual truncation constructs were expressed in transfected cells, but all had very different localisation patterns depending on the region of the NS3 protein expressed.

## The region between amino acids 67 to 100 within the Mouse Norovirus NS3 protein is responsible for translational shut off and apoptosis induction

To determine the region within the MNV NS3 protein that is responsible for translational shut off and apoptosis induction, we transfected 293T cells with full length WT NS3 (NS3-HIS), full length NS3 fused with mCherry and NS3-mCherry truncation mutants. At 23.5 hours post transfection puromycin was added to the cell, before cell lysates were collected for western blot analysis (Fig 5A and File Fig 1 at https://doi.org/10.26188/26332729) or fixed for FACS analysis (Fig 5B) and assessed with anti-puromycin antibodies. Our western blot and FACS analysis revealed that expression of MNV NS3-HIS and NS3-mCherry induced host protein translational shut-off, as determined by a loss of staining for the incorporated puromycin. However, the transfection and subsequent expression of NS3 encoding only amino acids 1 to 34 and 1 to 64 was not observed to shut-off translation (Fig 5A and 5B). Interestingly, in cells expressing MNV NS3 encoding amino acids 1 to 100, 1 to 134 and 1 to 182 host translational shut-off was once again observed. This indicates that the region within the MNV NS3 protein responsible for translational shut-off must be between amino acids 67 to 100. Further, when the core and C-terminal domain segments of the MNV NS3 protein (*i.e.* amino acids 135 to 300, 183 to 364 and 301 to 364) were expressed into 293T cells, the puromycin levels were similar to untransfected cells indicating not attenuation of host protein synthesis (Fig 5A and 5B).

To further the potential impact of the NS3 mutants on apoptosis induction, we transfected and expressed the MNV NS3 truncation mutants into 293T cells for 24 hours, collected cell lysates and assessed the cleavage and abundance of the apoptotic proteins PARP and MCL-1, respectively (Fig 5C). We observed that expression of full length NS3 and NS3 truncation mutants 1 to 100, 1 to 134 and 1 to 182 triggered the cleavage of PARP and a reduction in the amount of MCL-1 (Fig 5C). We did not observe any noticeable induction of the cleavage of PARP upon expression of the remaining NS3 mutants (Fig 5C). During expression of full length NS3 and the NS3 truncation mutants 1 to 100, 1 to 134 and 1 to 182, and upon addition of QVD, a pan-caspase inhibitor, and MG-132, a proteasome inhibitor, we observed a significant prevention in appearance of the cleavage product of PARP (Fig 5D) and a restoration in the reduction and abundance of MCL-1 (Fig 5E).

Again, these results indicate that the 67 to 100 amino acid region within MNV NS3 is not only responsible for the observed host translation shut-off, but also the induction of apoptosis via a reduction in MCL-1 levels and the associated cleavage of PARP to induce apoptosis.

## NS3 Alanine scanning reveals the exact amino acids responsible for translational shut-off and PARP cleavage

Alanine scanning is a site-directional mutagenesis technique used to identify the role of specific amino acids by changing the amino acid in question to an Alanine. In our attempts to identify the amino acids within the MNV NS3 protein that are responsible for translational shut off and apoptosis, we mutated three consecutive amino acids to Alanine's, starting at amino acids [70]Leu-[71]Leu-[72]Ser, and continued with the next 3 amino acids overlapping by 1 amino acid *i.e.* [72]Ser-[73]Asn-[74]Met (Primer list in File Table 2 at https://doi.org/10.26188/26332729). In the event of an Alanine presence, this was changed to a Glycine (Fig 6A and 6B).

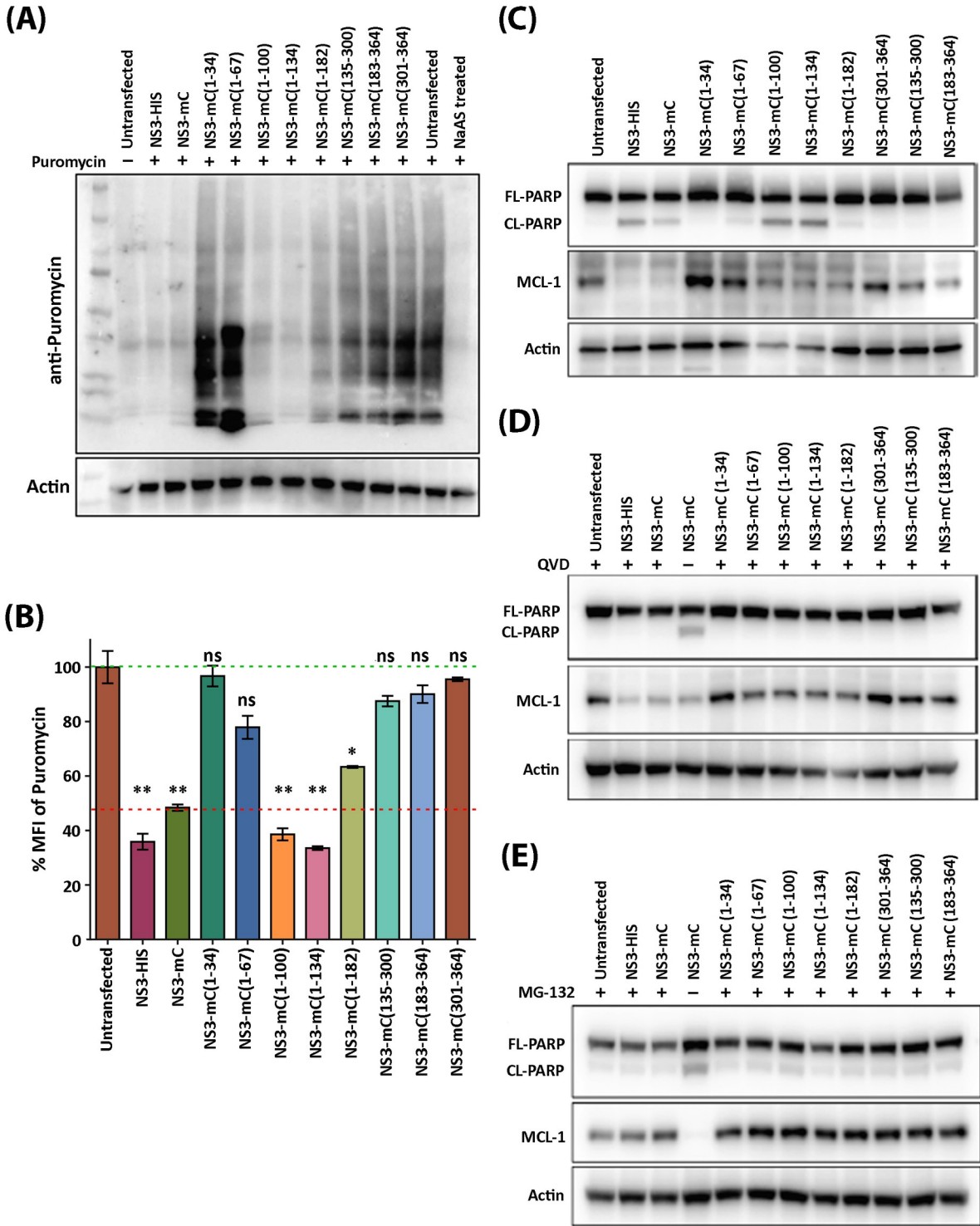

**Fig 5. N-terminal region of MNV NS3 induces translation repression and apoptosis.** 293T cells were transfected with expression plasmids encoding NS3 and truncation mutants for 24 hours. (A and B) Host cell translation was measured by treating with puromycin for 30 mins before (A) harvesting lysates and staining with anti-puromycin antibody by immuno-blot or (B) fixing cells and performing flow cytometry with anti-puromycin antibody. Mean fluorescent intensity (MFI) was normalised to untransfected cells. (C-E) immuno-blots were performed on lysates harvested from transfected cells following (C) no treatment (D) treatment with caspase inhibitor QVD or (E) treatment with proteosome inhibitor MG-132 for the duration of transfection.

We observed that expression of two out of the 15 mutations did not produce any protein due to unknown reasons (represented in Fig 6A and 6B by an asterisk).

Full length NS3, NS3 truncation mutants and NS3 Alanine mutants were transfected and expressed in 293T cells, and the cells were subsequently lysed and activation and cleavage of PARP was assessed by western blot analysis to determine apoptosis induction (Fig 6C) and stained for FACS analysis to determine the extent of host protein translation in the transfected cells (Fig 6D). As we had observed previously, NS3-mCherry (1 to 34) and NS3-mCherry (1 to 67) did not induce the cleavage of PARP, however expression of NS3-mCherry (1 to 100) was observed to induce the cleavage of PARP. In addition, NS3(70, 71, 72)-mCherry and NS3(74, 75, 76)-mCherry did not cause PARP cleavage (Fig 6C) or host translational shut-off (Fig 6D), indicating that these amino acids are vital for NS3's functions to shut-off translation and activate apoptosis. Interestingly, the Alanine mutant between these two regions, NS3(72, 73, 74)-mCherry, retained its activity to cleave PARP and shut-off translation (Fig 6C and 6D). We found that overall, mutating the second segment of NS3 (*i.e.* amino acids 80 to 100) did not prevent the NS3 mediated shut-off of protein translation, however mutation of certain amino acids from the first segment of NS3 (namely amino acids 70 to 80) prevented the ability of NS3 to shut off host protein translation, as assessed by puromycin incorporation (Fig 6D).

Overall, these results indicate that the region within the MNV NS3 protein encompassing amino acids 70–80 contains the functional attributes that result in host cell protein translation shut-down, ultimately driving the cleavage of PARP.

## Human norovirus NS3 protein induces apoptosis and shuts-off translation

Although MNV is used as a useful model to elucidate norovirus replication and pathogenesis, we additionally aimed to determine whether the human norovirus NS3 (HuNoV NS3) protein was similar to the MNV NS3 protein in inducing a shut off of host translation and thus apoptosis induction. Alignment analysis revealed that the MNV NS3 and HuNoV NS3 proteins were 54.8% identical at the amino acid level (Fig 7A, straight line) and 17.5% conserved (Fig 7A, two dots). The identical and conserved regions were mainly clustered to the core and C-terminal regions of both proteins, with the N-terminal region showing higher amounts of disparities. The region we had identified in MNV NS3 that was attributable to the shut off of host translation and activated apoptosis (NS3 amino acids 68 to 100) (Fig 7A, Red segment) displayed a low level of identical (28.1%) and conserved (37.5%) amino acids with HuNoV NS3.

To determine if HuNoV NS3 could induce translational shut-off and apoptosis activation, we treated untransfected, MNV NS3 and HuNoV NS3 transfected 293T cells with puromycin for 20 mins, harvested lysates and analysed the samples by western blotting. We immuno-labelled with anti-puromycin antibodies to determine translation levels and anti-PARP antibodies to determine apoptosis activation by detecting full-length and cleaved PARP (Fig 7B). Our western blot analysis revealed that HuNoV NS3, similar to mouse NS3, reduced puromycin incorporation, indicating that host-translation is attenuated (Fig 7B). In addition, we observed that expression of both 6xHis- and mCherry-tagged HuNoV NS3 induced the cleavage of PARP resulting in apoptosis activation (Fig 7B). Additionally, we performed immuno-fluorescence (IF) analysis/microscopy by transfecting Hela cells with HuNoV NS3-mCherry and mouse NS3-mCherry, treated with Puromycin for 20 mins, fixed, visualised for mCherry expression and immune-labelled with anti-puromycin antibodies (Fig 7C). The IF analysis also showed that expression of both HuNoV NS3 and MNV NS3 transfected cells (Fig 7Ce and 7Ch) displayed a reduced level of Puromycin incorporation when compared with the untransfected bystander cells (Fig 7Cd and 7Cg), which was also supported by quantitative analysis (Fig 7D).

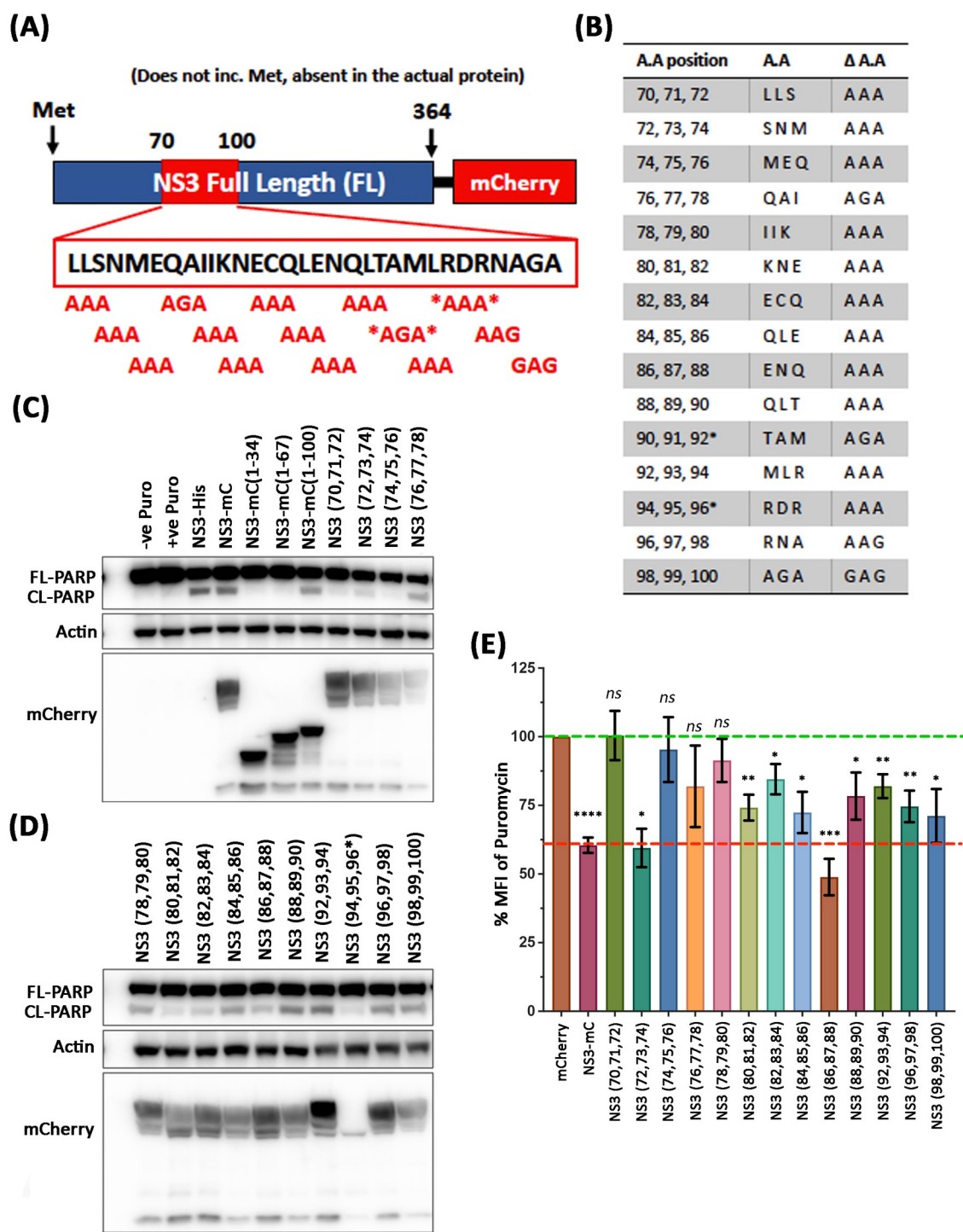

**Fig 6. Identification of key NS3 residues for translational shut-off and apoptosis.** (A) Schematic of mCherry-tagged NS3 tripartite alanine mutants generated in pcDNA3.1 expression constructs. Amino acid residues in region between 70 to 100 is illustrated in the expanded box along with changes in red. * denotes where generation of mutant was unsuccessful. (B) Table outlining positions of NS3 tripartite mutations, the original amino acids (A.A) and the specific change (Δ A.A). (C) Immuno-blot of lysates harvested 24 h.p.t. from 293T cells transfected with NS3 expression constructs encoding full length, truncation, or tripartite alanine mutant proteins. (D) 293T cells transfected with full length NS3 or tripartite alanine mutant constructs for 24 hours were treated with puromycin for 30 mins before fixation and flow cytometry was performed. Mean fluorescent intensity (MFI) of puromycin staining was normalised to control cells transfected with mCherry only.

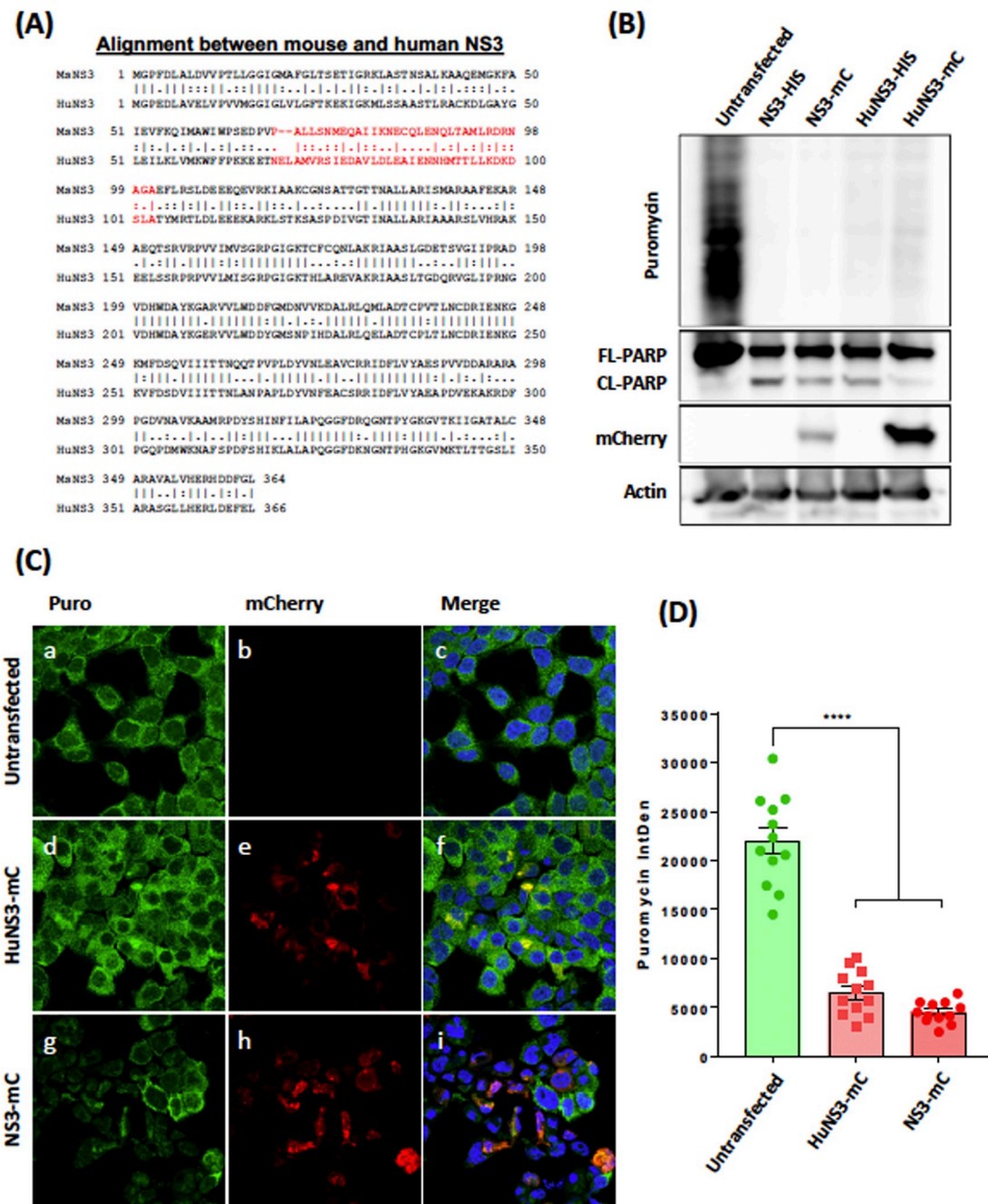

**Fig 7. HuNoV NS3 induces translation repression and apoptosis.** (A) Amino acid homology between MNV NS3 and HuNoV NS3. Alignment generated with EMBOSS Needle tool using Needleman-Wunsch algorithm. MNV NS3 amino acids 68 to 100 have been highlighted in red (B) Immuno-blot of lysates transfected with expression plasmids encoding MNV NS3 or HuNoV NS3 fused with either His Tag or mCherry for 24 hours. Cells were treated with puromycin 30 mins before harvest and translational assessed by measuring incorporation with anti-puromycin antibody. (C) Immunofluorescence was performed on Hela cells transfected with either HuNoV NS3, MNV NS3 or left untreated and treated with puromycin 30 mins before fixation. Cells were stained with DAPI (blue) anti-puromycin (green) and mCherry-tagged HuNoV or MNV NS3 visible (red). (D) Fluorescence intensity was quantified in mCherry positive cells relative to untransfected bystanders. Images were captured using a Zeiss LSM 710 confocal microscope and analysed with ZEN software.

Overall, these results indicate that the functional capacity of the norovirus NS3 protein to repress host cell protein translation and thereby induce the programmed cell death pathway of apoptosis is conserved between murine and human viruses.

## Structural modelling of the MNV NS3 protein with and without introduced mutations

In attempts to understand how the mutations introduced into the MNV NS3 could affect the functionality of the protein we used structural modelling to observe any significant changes in the predicted 3D model of the protein using AlphaFold2 (Fig 8). Our predicted modelling of full-length WT NS3 is in agreement of the recently described model for the N-terminus of NS3 although we can also resolve 3 α-helices in a bundle, not 4 at the N-terminus. This could be explained by the fact that we have modelled the full-length protein and not just the truncated N-terminus. In addition to the WT protein we also modelled our NS3(70, 71, 72) mutant protein that is unable to suppress host protein translation and thus apoptosis. Here we can observe a very similar structure of the protein, except for a dramatic change in the N-terminus of the protein whereupon only a singular and linear α-helix is observed.

Based on the recent report [35] this structural change would significantly disrupt the formation of potential pore-forming domains and provide a structural explanation why NS3(70, 71, 72) has reduced functional capability. This structural deformity would also perturb any potential protein-protein or protein-lipid interactions that define translational repression.

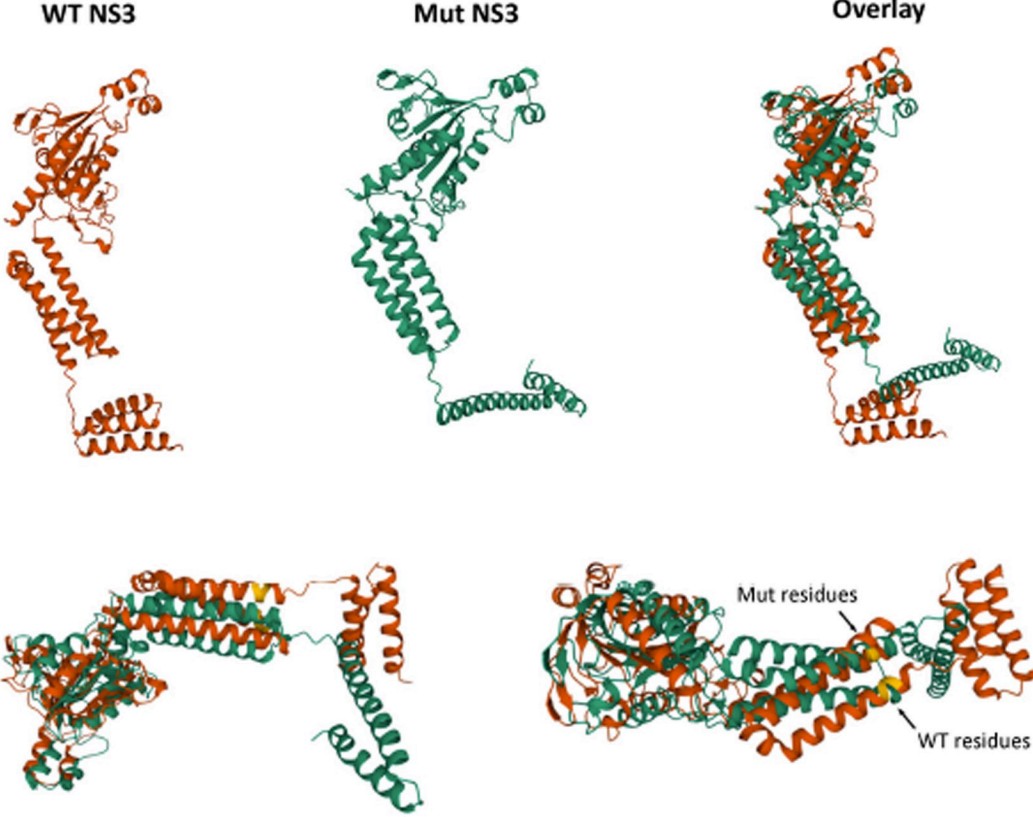

**Fig 8. Predictive modelling of the WT and mutant MNV NS3 protein.** The sequence of the WT CW1 NS3 protein (orange) and NS3(70,71,72) (green) protein were imported into AlphaFold2 to derive a predicted model of the proteins. The models were overlaid to indicate the changes in protein structure upon mutation and the amino acids mutated are indicated in yellow.

## Discussion

Apoptosis is a form of non-inflammatory programmed cell death that serves a key role in the innate immune response and clearance of many invading pathogens. On the other hand, some pathogens have hijacked this response for the benefit of their replication and evolved mechanisms to manipulate the cell death pathways. For MNV, apoptosis has been shown to be essential for efficient replication of the virus in cell culture and required for caspase-3 dependent proteolytic cleavage of the NS1/2 proteins [46]. Recent studies have shown that processing of NS1/2 has a number of important implications on MNV replication *in vivo* including; Essential for intestinal epithelial cell tropism and Tuft-Cell infection [47,48]; resistance to interferon-λ [47]; persistent shedding with some MNV strains [34]; and amplification of apoptosis [34]. While the requirement for apoptosis during norovirus infection is recognised, the mechanism of induction is not well understood, nor the viral proteins involved. In this study we have described a previously unrecognised mechanism for the induction of intrinsic apoptosis during MNV infection.

As discussed above, apoptotic cell death is known as an important hallmark of MNV infection. During our studies we observed an accumulation of virus in the cellular supernatant that was tightly associated with loss of cell viability and cleavage of apoptosis markers PARP and caspase-3 from 15 h.p.i.. This supports findings from us and others that apoptosis-mediated cell death occurs during MNV infection and is potentially facilitating replication. We also observed that over-expression of the viral protein NS3 was sufficient to induce apoptosis independent of replication. Importantly, this suggests that MNV actively induces cell death to benefit propagation compared with an indiscriminate consequence of infection. Only one recent study has linked expression of the MNV NS3 protein to cell death, which proposed that NS3 potentially forms pores in the mitochondria membrane [35], whilst another study could not show this despite transfecting each non-structural expression plasmid independently, possibly due to using RAW264.7 cells with a low transfection efficiency [30]. In this study we suggest that NS3 expression results in the depletion of pro-survival proteins resulting in apoptosis induction; the same outcome but a different mechanism to that described above. Another report showed that expressing the entire ORF1, encoding all non-structural proteins, could induce apoptosis [33]. While we saw no evidence of cell death or apoptosis in cells expressing the individual NS4, NS5, NS6, or NS7 proteins, we did observe cleavage of PARP in cells expressing the NS1/2 protein albeit to a lesser degree than in cells expressing NS3 alone (Fig 2). Caspase-3 cleavage of NS1/2 can potentiate apoptosis through an unknown mechanism [34] and while we did not explore this as part of this study, we did observe that co-expression of NS1-2 and NS3 (and mutants) did potentiate lytic cell death above expression of the single protein alone (File Fig 3 at https://doi.org/10.26188/26332729). Thus, the co-expression of NS1/2 and NS3 may appear to provide a synergistic role in the rapid induction of cell death (both lytic and non-lytic) during infection. An aspect that we aim to pursue further.

The identification of NS3 involvement in MNV-induced cell death was of particular interest due to our recent findings that NS3 induces repression of host cell translation [21]. In this study we again showed that over-expression of the MNV NS3 protein could drastically limit cellular translation and show that this occurs independently of NS3-mediated apoptosis induction (Fig 2). Translation repression has a well-established link to apoptosis induction through the loss of a key pro-survival protein, MCL-1 [49]. It has been observed for a bacteria and viruses that restricting protein synthesis prevents the ongoing replenishment of MCL-1 which has a short half-life and is rapidly turned-over under normal homeostatic conditions. It has been shown that infection with some viruses results in either stabilisation or depletion of MCL-1 to regulate apoptosis [50]. For SARS-CoV-2 it was shown that expression of the N

protein binds MCL-1 to prevent apoptosis and promote virus replication [51], whereas infection with VSV results in depletion of MCL-1, subsequent induction of cell death programs and restricts virus replication [52]. Herein we show that either infection with MNV or expression of the NS3 protein alone, resulted in depletion of MCL-1 leading to apoptosis induction (Figs 1 and 2). Proteasomal inhibition could rescue MCL-1 levels during NS3 expression and prevent apoptosis induction (Fig 2), further supporting our hypothesised model that NS3-translation repression results in MCL-1 depletion and intrinsic apoptosis. This represents a new mechanism for induction of apoptosis not previously observed during norovirus infection and a key finding in our understanding of norovirus replication.

Modulation of the apoptotic pathway by MNV is not limited to the mechanism described in our study. Previous reports have outlined three modulatory mechanisms which presumably operate together to control cell death during infection. Survivin is an anti-apoptotic member of the IAP group of proteins and while recent reports have called into question the mechanism of survivin, it is widely believed that this protein works directly and with X-IAP to inhibit caspases activated during apoptosis [53,54]. MNV has been reported to downregulate the transcription of BIRC5, the gene encoding survivin to promote apoptosis [30]. Because the targets of survivin are downstream from mitochondrial permeabilization it is hypothesised that this acts as a supplementary system to promote cell death during infection. Lysosome-associated cysteine protease cathepsin-B has also been identified as activated during MNV infection and implicated as a possible contributor to apoptosis [32]. While this finding has not been extensively assessed, during lysosome disruption, cathepsin B can induce apoptosis [55], however, it remains unclear if this noncanonical pathway is connected to the mechanism for cell death we observed in this study. The MNV expressed protein VF1 may act as a modulator of apoptosis, with a report showing that a VF1-knockout virus had higher caspase activity than the wild-type [6]. We did not explore this connection during our study, but we propose that this may act in combination with NS3 to control apoptosis. However, we did observe some PARP cleavage upon expression of NS1/2 (Fig 2), and upon co-expression of NS1/2 with NS3 or mutants we did observe some synergistic effects in promoting lytic cell death (File Fig 3 at https://doi.org/10.26188/26332729). This aspect is something we are currently exploring further. Finally, a potential mechanism has been proposed whereby the N-terminus of NS3 mimics an MLKL-like protein to promote pore formation on the mitochondrial membrane and thus invoke cell death [35]. Although we have not observed anything similar and our immunogold labelling appeared not to detect NS3 on the mitochondria through the course of infection (Fig 3) or during expression of the mutant proteins (File Fig 2 at https://doi.org/10.26188/26332729), it does not discount that NS3 mediates multiple functions in infected cells. We were clearly able to correlate the induction of cell death observed in infected murine macrophages with the functional capacity of expressed NS3.

In addition to identifying the viral NS3 protein as responsible for translation repression and apoptosis induction, we were also able to expand on our previous findings and identify a key domain involved within the N-terminal domain of the protein (Figs 5 and 6). Encouragingly the region of NS3 identified by us supports recent findings that identified the same region using an overexpression system of MNV NS3 and the human norovirus NS3 homologue [15,16,35]. Their findings also recognise the distinct vesicle localisation of mutants able to induce apoptosis consistent with our findings and determined these to be lipid droplets [15]. It is not clear what the significance of lipid droplet localisation is for translation repression and apoptosis. The similarity in domain and localisation patterns between our MNV NS3 study and those published on HuNoV NS3 clearly suggests a shared pathway between both viruses. This is further supported by our observations that the HuNoV homologue also represses host cell translation resulting in apoptosis (Fig 7).

Together this study has shown that host cellular translation is repressed by a key region in the N-terminal domain of both human and MNV NS3 leading to a dysregulation of MCL-1 and the induction of intrinsic apoptosis. Given the importance of NS3 in evading the immune response and the essential requirement of apoptosis on viral replication this key discovery opens the exciting possibility of rational drug design and development of attenuated live-virus vaccines.

## Acknowledgments

We wish to thank Kim Green (NIH) for kindly donating the anti-NS3 antibodies. The authors acknowledge the facilities, and the scientific and technical assistance, of the Australian Microscopy & Microanalysis Research Facility at the Centre for Microscopy and Microanalysis, The University of Queensland, and the University of Melbourne's Biological Optical Microscopy Platform (BOMP).

## Author Contributions

**Conceptualization:** Turgut E. Aktepe, Joshua M. Deerain, Jaclyn S. Pearson, Peter A. White, Jason M. Mackenzie.

**Data curation:** Turgut E. Aktepe, Joshua M. Deerain, Jennifer L. Hyde, Svenja Fritzlar, Eleanor M. Mead, Julio Carrera Montoya.

**Formal analysis:** Turgut E. Aktepe, Joshua M. Deerain, Jennifer L. Hyde, Svenja Fritzlar, Eleanor M. Mead, Julio Carrera Montoya, Jason M. Mackenzie.

**Funding acquisition:** Peter A. White, Jason M. Mackenzie.

**Investigation:** Turgut E. Aktepe, Joshua M. Deerain.

**Methodology:** Joshua M. Deerain, Jennifer L. Hyde, Svenja Fritzlar, Jaclyn S. Pearson.

**Project administration:** Jaclyn S. Pearson, Peter A. White, Jason M. Mackenzie.

**Resources:** Jennifer L. Hyde, Jaclyn S. Pearson, Jason M. Mackenzie.

**Supervision:** Jaclyn S. Pearson, Peter A. White, Jason M. Mackenzie.

**Validation:** Turgut E. Aktepe, Joshua M. Deerain.

**Writing – original draft:** Turgut E. Aktepe, Joshua M. Deerain, Jason M. Mackenzie.

**Writing – review & editing:** Turgut E. Aktepe, Joshua M. Deerain, Svenja Fritzlar, Abderrahman Hachani, Jaclyn S. Pearson, Peter A. White, Jason M. Mackenzie.

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
