## [Decision Letter · Decision Letter 0]

22 Feb 2024

Dear Prof Mackenzie,

Thank you very much for submitting your manuscript "Norovirus-mediated translation repression promotes macrophage cell death" for consideration at PLOS Pathogens. As with all papers reviewed by the journal, your manuscript was reviewed by members of the editorial board and by several independent reviewers. In light of the reviews (below this email), we would like to invite the resubmission of a significantly-revised version that takes into account the reviewers' comments.

Overall the opinions on your manuscript were split and several substantial concerns were raised. Reviewer 1 raises major concerns on novelty and impact, which are not shared by the other reviewers. However, point 2 and 3 suggest critical experimentation, including a complementation experiment and a more rigorous view on the (lack of) mitochondrial co-localization, that need to be addressed in a revision. The comments of reviewer 2 mainly address the potential physiological relevance of the findings, regarding the role of apoptosis for particle release and the function of NS3 in translational inhibition. The authors should particularly pay attention to linking the identified phenotypes better to the viral life cycle by adding experiments also in other cell types, thereby better demonstrating the importance of NS3 functions in apoptosis and translational repression on virial replication and release. 

In addition, all reviewers ask for a number of editorial changes, including the implementation of critical literature that need to be addressed.

We cannot make any decision about publication until we have seen the revised manuscript and your response to the reviewers' comments. Your revised manuscript is also likely to be sent to reviewers for further evaluation.

Sincerely,

Volker Lohmann

Guest Editor

PLOS Pathogens

Ashley St. John

Section Editor

PLOS Pathogens

Michael Malim

Editor-in-Chief

PLOS Pathogens

orcid.org/0000-0002-7699-2064

Overall the opinions on your manuscript were split and several substantial concerns were raised. Reviewer 1 raises major concerns on novelty and impact, which are not shared by the other reviewers. However, point 2 and 3 suggest critical experimentation, including a complementation experiment and a more rigorous view on the (lack of) mitochondrial co-localization, that need to be addressed in a revision. The comments of reviewer 2 mainly address the potential physiological relevance of the findings, regarding the role of apoptosis for particle release and the function of NS3 in translational inhibition. The authors should particularly pay attention to linking the identified phenotypes better to the viral life cycle by adding experiments also in other cell types, thereby better demonstrating the importance of NS3 functions in apoptosis and translational repression on virial replication and release. 

In addition, all reviewers ask for a number of editorial changes, including the implementation of critical literature that need to be addressed.

Reviewer's Responses to Questions

**Part I - Summary**

Reviewer #1: The manuscript by Aktepe et al describes the manipulation of intrinsic apoptosic by Murine Norovirus and the role of a specific domain within the NS3 protein in this process.

Overall the authors show convincingly that NS3 is important for promoting cell death pathways through the translational silencing of the MCL-1 protein, a key repressor of intrinsic apoptosis. Using deletions and mutational analysis they further map the domain and residues within NS3 that are responsible for this effect. The results are solid and self-explanatory with little room for alternative explanations and conducted in a rationale manner. However, to this reviewer, the study overall lacks depth and novelty. Previous work from Wang G et al 2023, Yen JB 2018 and 2021, had already investigated the regulation of cell death by noroviruses and identified specific proteins involved in this process. Therefore, rather than providing real novelty, the current study feels rather incremental in nature and one-dimensional, being mostly based on the repetition of similar assays using deletions mutations and/or point mutations. Therefore, it does not reach the level this reviewer would expect for publication in a journal of P.PATH standing.

Beyond this general assessment, there are a few specific points that the reviewer would like to see addressed as outlined below.

Reviewer #2: MacKenzie et al., identify a Murine Norovirus nonstructural protein NS3 to be critical for shutting down host translation and inducing apoptosis/cell lysis/cell death.

Their conclusions are based largely on ectopic expression of NS3 in cells and assays measuring host translation and apoptosis.

While the study is intriguing, it is not convincing that this "role" they have found for NS3 is of relevance in the lifecycle of the virus across different cell types.

Major concerns

All the data in the paper connecting NS3 to protein synthesis effects is with ectopic expression of the NS3 alone in cells. The authors do not have any data showing that NS3 is activating apoptosis and shutting down translation in the context of the viral infection . How does NS3 effects observed reconcile with PKR- which is known to be rapidly activated and shut down host protein synthesis when viral RNA is detected - no data on this, not even discussion. Thus the authors findings just come out of no where, no context.

Figure 1 is problematic: Authors plotted Figure 1A to start at 9hrs post infection of their cultures.

At 9hrs post infection, there is already ~106 PFU/ml virus having been released into the supernatant with only 10% drop in "cell viability". Secondly the way the authors measure cell viability is by looking at the supernatant for cytoplasmic markers- but we know that MNV can be released in exosomes, so their viability measurements are likely taking into account the exosomal cytosolic contents.

The authors need to actually measure cell permeability directly.

By 15 hrs post infection, there is already ~108 PFU/ml virus released into the supernatant, with only 30-40% drop in "cell viability" ( based on the error bars)

So 6 logs of virus have come out of these cells with only a 10% drop in "cell viability". This completely undermines the authors premise that the loss of cell viability is needed for virus release.

Indeed a previous study ( Cell Host Microbe 2018) had shown in RAW cells that cell death was not needed for MNV-1 release ( virus was released in exosomes).

More discrepancies : In Figure 1C- NS7 does not show up in the blot until 12 hours post infection, yet in Figure 1A we already see >~106 PFU/ml of newly replicated assembled virus released into the supernatant.

Without NS7 there would not be virus made- so there is something askew about these two figures.

Furthermore, Cleaved PARP and CASP3 do not appear until 15 hours post infection- - yet the first 15 hours we have nearly ~108 PFU/ml virus released

So apoptosis is not a prerequisite for virus release- by any measure-

Authors need to reanalyze their figures and modify the text so that it faithfully represents their actual data.

Authors should also check their findings in other cell types.

Reviewer #3: The authors have delineated a novel mechanism by which murine and human noroviruses induce intrinsic apoptosis through NS3-mediated translation repression. This discovery holds significant relevance not only within the norovirus field but also more broadly in understanding the role of apoptosis in viral infections. Specifically, the authors demonstrate that murine norovirus (MNV) actively induces cell death to promote its propagation. While this study represents the second to link NS3 with apoptosis, it is the first to associate this effect with a narrow domain at the N-terminus of NS3. Building upon their prior work describing translational shutdown for murine noroviruses, this manuscript provides additional insights into the molecular mechanisms underlying viral pathogenesis. In addition to MNV, in vitro studies utilizing recombinantly expressed NS3 suggest a similar phenomenon in human norovirus, despite minimal sequence homology at the respective site.

The manuscript is well-crafted, featuring a comprehensive introduction, innovative methodologies, and stimulating discussions. Experiments were conducted across diverse cell lines, including immortalized primary bone marrow-derived macrophages (iBMDM), and various in vitro systems with modified proteins to enhance detection. The combined data are consistent and conclusive, and the conclusions drawn are well-justified.

**Part II – Major Issues: Key Experiments Required for Acceptance**

Reviewer #1: First, the introduction and discussion of existing literature is highly biased and completed omits several key studies for other investigators who have dissected in great detail the regulation of translation during norovirus infection. Specifically, line 50 when discussing interaction of VPg and norovirus proteins with translation factors, the authors should discuss findings from Emmott et al Mol Cell Proteomics 2018 and Hosmillo et al 2020 eLife. Later line 59, the authors refer to the overall regulation of translation and stress granules by MNV without acknowledging studies from Brocard et al 2020 P.Path that have addressed this topic. Line 70, the authors comment on specific regulation of translation by MNV and gaps in knowledge without commenting on the known cleavage of PABP by MNV NS6 (Emmott et al 2018) and regulation of p38 signalling by MNV1 (Royal E et al JBC 2015). Overall the authors are encouraged to be less targeted in their discussion of the known literature of translational control during MNV infection.

Second, the authors identify a single helical domain important for NS3 functions and the regulation of apoptosis. Given what is already known – as the authors discuss later in the discussion – the authors should perform complementation experiment whereby the DNS£ construct are supplemented with either NS1/2 or VF1 to evaluate the redundancy and synergistic effects of several norovirus proteins in regulating virus-induced cell death. Can the expression of these proteins rescue the loss of NS3 function in the regulation of apoptosis and translation?

Third, the authors were not able to recapitulate mitochondrial localisation of NS3 during infection or overexpression of NS3 yet comment on the potential impact of the domain identified in regulating the pore-inducing ability of NS3. The authors should explore this further and provide evidence that the loss the alpha-helix they identify is or isn’t required to regulate the localisation to mitochondrial membrane (perhaps using various cell lines or primary cells) and pore formation to regulate mitochondrial function.

Reviewer #2: (No Response)

Reviewer #3: (No Response)

**Part III – Minor Issues: Editorial and Data Presentation Modifications**

Reviewer #1: in the discussion the authors comment on the regulation of MCL-1 translation during norovirus infection. Given the existing knowledge of MNV-induced transcriptional and translational regulation on genome-wide level (transcriptomics, translatomics and proteomics) could the authors comment on whether this study had identified MCL-1 as specifically translationally regulated during infection?

Reviewer #2: (No Response)

Reviewer #3: Minor comments include requests for the indication of kDa sizes on immunoblots, correction of missing information in figure legends, abbreviation consistency and additional technical background for Figure 3.

Specific issues raised in the manuscript include inconsistencies in abbreviation usage, clarity on antibody specificity for human and mouse proteins, and some optional suggestions for further analyses to strengthen the findings.

Introduction: none

Materials and Methods:

- it was not clear what the guinea pig ati-NS3 AB was used for (M&M and acknowledgements), from the figures it appears that immunofluorescence was performed using mCherry only

- since you switch between human and mouse cell lines please indicate which antibodies recognize human and mouse proteins

#99: IBMM = IBMDM use of abbreviation is inconsistent, change in text and figure legends

Results:

General

The abbreviation of HuNS3 is confusing and not consistent with the MNV notation “MNV NS3” mayby “HuNV NS3”

#242/243: Please give a reference on using the puromycin incorporation assay as a marker for translation shut off maybe Hsu 2021 ?

#300 - The N-terminal constructs 1-14 and 1-67 are intruguing but the interpretation seems only applicabale to some cells and this should be mentioned.

- in line with the truncated NS3 data from figure 5 one should note here that FL, 1-100/134/182 have similar cytoplasmic localization

#320 - "to further these studies": please be specific what hypothesis your are addressing

#359 ..amino acids 70-80.. "ultimately driving cellular apoptosis via a reduction in the pro-survival BCL-2 family of proteins." only PARP cleavage is shown not the pro survivial BCL-2 family proteins like MCL-1, BCL-XL or BCL-2, please clarify

Discussion:

- Maybe highlight the role of translational shut off and Mcl-1 downregulation in other viral infections giving examples, particularly with reference when a puromycin incorporation assay was used

#409: replication of virus in vitro, do you mean in cell culture please clarify

#425: please discuss the differences of this study and ref 31

#431: NS1-2 shows a double band in figure 2a this could indicate that there is some cleavage into NS1 and NS2? There also seems a double band for NS3.

#465: NS3-mcherry fusion may impact protein structure and thereby cellular localication and needs to be taken into account when discussing localization

Fig1:

Figure heading: should include the celltype “iBMDM”

(c) abbreviations FL and CL should be spelled out in the legend once

(d) quantification from immuno-blots should be explained how this was done. If this was performed using image based quantification it should be indicated that this is an estimation

Fig2:

there is no asterisk as indicated in the figure legend

(a) maybe name this puromycin incorporation assay rather than immun-blot

there is no band for NS4 abd NS6 is barely visible??

please define mock, -puro and -ve control

please include the kDa to the immunoblot

(b) Missing text in legend, (b) not explained; what is NaNS

(c) what happens to caspase3 cleavage here?

a better control to mock may have been DMSO?

Fig3:

hrs p.i. is not consistent

how is the RC defined ?

d), e) and f) are missing in the legend

it is not explained how the cytoplasmic distribution was quantified

Fig 4:

There is no yellow area for TM in the figure

There is no “staining“ but mCherry fluorescence

Fig 8:

- maybe this can be moved in the appendix as these are just predictions and not data

- have you considered Linear and helical wheel projections of helix A of the mouse and human model particularly of helix 3 highlighting the mutations at positions 70-80

Fig S1:

- can be omitted

Figure Quality

The resolution of figures in the provided manuscript was terrible and the text in the figures was barely readable, the BioRxiv PDF had better quality than the provided manuscript

Future directions/discussions may include investigating whether NS3 from non-acute/chronic strains can induce apoptosis and exploring compensatory mechanisms for impaired apoptosis, such as non-apoptosis-mediated release of virus through vesicles.

PLOS authors have the option to publish the peer review history of their article (what does this mean?). If published, this will include your full peer review and any attached files.

Reviewer #1: No

Reviewer #2: No

Reviewer #3: No
---

## [Decision Letter · Decision Letter 1]

5 Aug 2024

Dear Prof Mackenzie,

We are pleased to inform you that your manuscript 'Norovirus-mediated translation repression promotes macrophage cell death' has been provisionally accepted for publication in PLOS Pathogens.

Best regards,

Volker Lohmann

Guest Editor

PLOS Pathogens

Ashley St. John

Section Editor

PLOS Pathogens

Michael Malim

Editor-in-Chief

PLOS Pathogens

orcid.org/0000-0002-7699-2064

Reviewer Comments (if any, and for reference):

Reviewer's Responses to Questions

**Part I - Summary**

Reviewer #2: Reviewer would like to thank the authors for responding thoughtfully to previous comments and modifying to the text. Whether there is a role of apoptosis for MNV replication and egress or its a side effect of viral proteins made still remains to be understood. It is unfortunate that the authors could not provide data in the context of viral infection ( rather than ectopic expression) but this may not be technically possible if such mutations disrupt fundamental aspects of the viral replication.

**Part II – Major Issues: Key Experiments Required for Acceptance**

Reviewer #2: (No Response)

**Part III – Minor Issues: Editorial and Data Presentation Modifications**

Reviewer #2: (No Response)

PLOS authors have the option to publish the peer review history of their article (what does this mean?). If published, this will include your full peer review and any attached files.

Reviewer #2: No

---

## [Editor Report · Acceptance letter]

28 Aug 2024

Dear Prof Mackenzie,

We are delighted to inform you that your manuscript, "Norovirus-mediated translation repression promotes macrophage cell death," has been formally accepted for publication in PLOS Pathogens.

Best regards,

Michael Malim

Editor-in-Chief

PLOS Pathogens

orcid.org/0000-0002-7699-2064